# Investigation of Different Types of Micro- and Nanostructured Materials for Bone Grafting Application

**DOI:** 10.3390/nano12213752

**Published:** 2022-10-25

**Authors:** Sara Targonska, Sebastian Dominiak, Rafał J. Wiglusz, Marzena Dominiak

**Affiliations:** 1Institute of Low Temperature and Structure Research, Polish Academy of Sciences, 50-950 Wroclaw, Poland; 2Department of Oral Surgery Medical, University of Wroclaw, 50-425 Wroclaw, Poland

**Keywords:** bone grafting, bone reconstruction, autograft, allograft, xenograft, nanostructure, bone substitution

## Abstract

The insufficient volume of dental ridges is one of the most severe problems regarding an oral cavity. An inadequate amount can cause problems during various types of dental treatment. Its complexity originates from the etiopathogenesis of this problem. In this study, the representatives of auto-, allo-, and xenografts are compared. The physic-chemical differences between each of them were evaluated using XRD (X-ray Powder Diffraction), a SEM (Scanning Electron Microscopy), FT-IR (Fourier transformed infrared spectroscopy), and TGA (thermogravimetric analysis). Based on the SEM images, it was observed that the origin of the material has an influence on collagen fiber compact level and porosity. Following a comparison of FT-IR spectra and XRD, the crystal and chemical structures were described. Based on TGA, different water concentrations of the investigated materials, their high thermal stability, and concentration of inorganic phase, hydroxyapatite was determined. The presented study is important because it delivers information about chemical structure and its impact on bone regeneration. This knowledge should be taken into consideration by dental clinicians, because different types of bone grafts can accommodate the achievement of various goals.

## 1. Introduction

The inadequate bone volume of the alveolar process of the maxilla and the alveolar part of the mandible can be found in the majority of patients undergoing various types of dental treatment [1,2]. It can not only complicate implantation procedures, but also prosthodontic ones, or even disable orthodontic treatment. The most common is tooth loss or various types of traumata. Moreover, it can occur as congenital anomalies related to micrognathia, hypodontia, or inflammatory processes [1]. To successfully treat patients in such conditions, it is necessary to regenerate bone defects. This can be accomplished by augmentation procedures [3].

An ideal grafting material should possess optimal properties, such as: osteoconductivity, osteoinducity, and potential for osteogenesis. Osteoconduction is the ability to create suitable conditions and space for new bone formation. In other words, the augmented bone substitute acts as a scaffold for the healing process. Osteoinduction refers to the stimulation of bone formation related to the recruitment of undifferentiated mesenchymal cells to osteoblasts. Researchers have shown that the leading role in this process belongs to BMPs and growth factors. The osteogenic properties mean that the graft is able to promote bone growth, caused solely by its own living osteoblasts and osteocytes [4,5].

Various types of bone grafting materials are used to gain a sufficient volume of alveolar arches and the best quality of regenerated bone. In reconstructive oral surgery, such materials as autogenous, allogenous, and xenogenous bone or alloplastic substances are used [6]. An autogenous bone, a graft harvested from the patient, is still referred to as the golden standard. Not only does it have osteoconductive and osteoinductive properties, but it also has potential for osteogenesis [7,8]. Moreover, there is no risk of graft rejection due to its origin, or possible inflammation because of pathogens. However, there are also disadvantages, like the need for a second surgical site for bone harvesting, such as the chin, external oblique line, or extraorally iliac crest tibia or calvarial bone [9], or the finite quantity of the harvested material [10]. It is worth mentioning that there is another way to gain autologous grafting material. It can be obtained from extracted tooth grinding. Following Y. Kim’s study, a grinded tooth also has BMPs and collagen type I fibers, and these tooth-derived proteins are as reliable as those gained from bone [11]. Following Jepsen et al. [3] and their discussion based on four different studies, all of the different types of bone grafts are reliable in long-term observation. Not only is the residual bone enlarged, but it also allows continuation with prosthodontic treatment based on dental implants.

In the last decade, a new type of autogenous material—a processed extracted tooth—has been developed. It is especially attractive for oral surgery. In contrast to others, this material is prepared just before usage. It is a unique procedure which allows the extracted tooth to be useful once again. Unfortunately, not all teeth can be used. The teeth with endodontic fillings are disqualified, and all restorative fillings must be removed before grinding. Thanks to its granule-shape, it can be used in various situations, from the simplest tooth socket augmentation after extraction, to the complex alveolar ridge regenerations or sinus-lift procedures.

Allogenous bone is an augmenting material obtained from different individuals of the same species; in most cases, from cadavers. In comparison to autologous bone, it has no osteogenic potential due to sterilization and deproteinizing methods. Nevertheless, thanks to residual growth factors which are present on allografts, it has osteoinductive potential [12]. Its availability without another large operating site and being ready-to-use just after opening are other advantages which have contributed to the popularity of this material among oral surgeons. On the other hand, using allogenic bone leads to some problems, such as possible transmission of viruses and bacteria [5].

Xenogenous bone grafts share just one property of an ideal bone substitute; namely, osteoconductivity. Because of their bovine and equine origin, they must undergo complete and strict sterilization procedures. That is why their usage is limited to maintaining space, or mixed with other osteoinductive materials for augmenting procedures [13,14]. Similarly to the allogenous bone, xenogenous bone can be used in large quantities with no harm to the patient. Xenografts are well known for long resorption times, which can be either useful or problematic. In the case of a sinus lift and implant placement, it is a desirable feature; however, in teeth movements in orthodontic treatment, it seems to be an obstacle [15,16].

Alloplastic bone substitutes can be either synthetic or natural, and the most commonly used examples here are: bioceramics, calcium sulfate, calcium carbonate, hydroxyapatite, β-tricalcium phosphate, and β-tri-calcium phosphate [16]. In the case of these materials, osteogenic and osteoinductive properties are not expected [8]. Thanks to its synthetic origin, every abovementioned bone substitute weakness can be overcome, which creates potential for bacteria and virus transmission.

In recent years, the market for biomaterials has been rapidly expanding. The availability and similar properties of bone substitutes allow surgeons to choose and mix a variety of them to achieve their goals, determine the ideal amount of bone needed for reconstructions, minimum morbidity, and a short healing period. The main objective of this study was to assess differences between the representatives of the most used bone substitutes by an examination of their chemical, physical, and morphological features.

## 2. Results

### 2.1. Morphology Analysis

The surface topography of the investigated bone-reconstructing materials was depicted in the SEM images in Figure 1, Figure 2, Figure 3, Figure 4 and Figure 5. It was detected that the microstructure strongly depends on material origin. The densely compacted collagen fibers with incorporated inorganic molecules were revealed in the SEM images. 

The SEM images of Allograft-Zimmer are depicted in Figure 1. The Allograft-Zimmer material was obtained from the cancellous bone. A typical spongy structure with holes with a diameter in the range of 100–350 µm was detected. Contrary to the later described materials, the surface of Allograft-Zimmer was relatively smooth, and not a single collagen fiber was detected. Even the crashed part of the surface was corrugated but compact. 

The surface of cortical bone (Allograft-Katowice) is presented in Figure 2. Small holes were detected on the bone surface, and they were identified as osteocyte canaliculi holes (see Figure 2, left side). The average diameter was evaluated to be 38 µm. The bone surface in the close environment of the osteocyte canaliculi hole was smooth, and infrequent fissures were observed. Most parts of the bone area were built by strongly bonded collagen fibers. Microcracks, with an average width equal to 2 µm, were found across the entire surface. The insert in the bottom right (see Figure 2) shows the structure in the area of the damaged part. This part was built by layered particles. 

In Figure 3, a part of the commercial bone replacing material, Allograft-Biobank, is shown. There are two different microstructures of collagen fibers with incorporated inorganic particles, which built the Biobank material. Collagen fibers on the particle surface were arranged parallel to the plane (see the insert on the left-hand side of Figure 3). Based on the SEM images (see the top left insert, Figure 3), the average diameter of collagen fibers was determined to be equal to 138 µm. The broken part, as well as the inside of the investigated material, are depicted in the insert on the right-hand side. Similar to Allograft-Katowice, a series of layers of compacted collagen fibers were observed. 

Figure 4 presents parts of Processed-Tooth. After the preparation process, irregular particles with an average size in the range of 130–600 µm were found. Across the surface there were gouges and grooves, probably formatted during the preparation by Kometa-Bio (see the insert on the left-hand side of Figure 4). Contrary to allograft-materials, on the surface of Processed-Tooth, asymmetric spherical-like shapes were formed. The surface showed high porosity. 

One representative sample of animal-bone-derived material, Geistlich Bio-Oss, is shown in Figure 5. A rough surface built by a mass of statically aggregated particles was observed. There was also a two-stage structure (see the insert on the left-hand side of Figure 5). The first stage was formed with particles of an average diameter equal to 0.353 µm. The small ones were collected and formed a structure with an average particle size equal to 1.395 µm. On Geistlich Bio-Oss, particles holes and pores generating a high surface were detected. 

### 2.2. Elements Content

The EDS analysis was performed to study the materials’ chemical composition. The individual elements content is listed in Table 1, and it is also graphically presented in Figure 6. The EDS analysis provides the information that, as expected, the bone grafting materials were composed mainly of calcium, phosphate, and oxygen atoms. It should be pointed out that the allograft materials (Allograft-Katowice, Allograft-Zimmer, and Allograft-Biobank) also included a noticeable amount of carbon: 31 mol%, 16 mol%, and 10 mol%, respectively. 

Moreover, the EDS analysis confirmed the presence of other elements; chlorine, magnesium, and sodium atoms also occurred. Allograft-Biobank included 3.6 mol% of sodium atoms. This was the highest concentration of minor elements in all the considered samples. The concentration of the other minor elements did not exceed 1 mol%.

### 2.3. Fourier-Transformed Infrared Spectra

The Fourier-transformed infrared spectra of the investigated bone-reconstructing materials are illustrated in Figure 7. The spectra are composed of a set of absorption peaks that are attributed to the organic as well as the inorganic phase included into bone structure. In the range between 2700–1900 cm^−1^, the spectra are not presented, because in this range energy absorption peaks were not detected. The broad band located between 3800–2810 cm^−1^ is associated with the presence of water molecules. The black dashed lines mark absorption peaks corresponding to the hydroxyapatite structure, especially to the phosphate groups. The lines ascribed to the ν_4_(PO_4_^3−^) triply-degenerated vibration modes of phosphate groups are located at 563 cm^−1^ and at 602 cm^−1^. The line at 961 cm^−1^ is assigned to the symmetric stretching modes ν_1_(PO_4_^3−^). Two peaks detected at 1042 cm^−1^ and at 1097 cm^−1^ are attributed to the antisymmetric stretching modes of phosphate groups ν_3_(PO_4_^3−^). In the spectra of Processed-Tooth and Geistlich Bio-Oss, weak and sharp peaks caused by stretching and bending vibrations of OH^−^ at 3573 cm^−1^ and at 632 cm^−1^, respectively, are detected [17,18].

In the crystal structure of hydroxyapatite, the OH^−^ and the PO_4_^3−^ groups can be substituted by the CO_2_^3−^ groups. The substitution of OH^−^ by the CO_2_^3−^ groups obtained a type A carbonate structure. If the carbonate ions are located at the phosphate groups position, the obtained structure is defined as type B carbonate [19]. Type AB carbonates are the structure with CO_2_^3−^ groups occurring in both positions. In the FT-IR spectra, two absorption lines located at 886 cm^−1^ and 1440 cm^−1^ are ascribed to the CO_2_^3−^ groups. The existence of these two bands suggests that the OH^−^ and the PO_4_^3−^ groups are substituted by the CO_2_^3−^ groups. Consequently, the HAp crystals present in all the investigated materials are indexed as type AB carbonates. The peaks corresponding to the CO_2_^3−^ groups are marked in Figure 7 with green borders. Carbon is also detected as symmetric and antisymmetric stretching bonds of -CH_2_ groups from collagen fibers. The -CH_2_ vibrational modes are observed at the FT-IR spectrum of Allograft-Katowice, mainly at 2976 cm^−1^ and at 2895 cm^−1^. Those vibrational modes are marked with a red border in Figure 7.

Other than the lines originating from the hydroxyapatite molecules, in the FT-IR spectra, lines corresponding to the organic component were detected. The organic phase mainly constitutes amide I-III from the protein molecules. The vibrations of amide I, amide II, and amide III were associated with the absorption peaks at around 1670 cm^−1^, 1550 cm^−1^, and at 1250 cm^−1^, respectively (see Figure 7, yellow frames). The lines originating from the amide I-III compounds are detected clearly in the spectra of Allograft-Zimmer and Allograft-Biobank. In the Allograft-Katowice spectra, weak absorption lines from amide I-III are found. Finally, in the spectra of both Geistlich Bio-Oss and Processed-Tooth, those lines were not detected [7,8]. 

### 2.4. Crystal Structure

The X-ray diffraction patterns of bone-reconstruction materials are illustrated in Figure 8. The detected patterns were composed of a broad peak in the range of small 2 theta angles and a series of narrow lines in the range of 20–60°. More intense sharp lines are detected at 2 theta 25.7° (002); 31.7° (211); 32.8° (300); 33.9° (202); 39.7° (310); 46.5° (222); and 49.3° (213). The diffraction peaks were completed with theoretical pattern ICSD no. 180315, and the pure hexagonal structure of hydroxyapatite crystals is found [4]. It is observed that the full width at half maximum (FWHM) is higher for Allograft-Biobank, Allograft-Zimmer, and Allograft-Katowice samples than for Geistlich Bio-Oss and Processed-Tooth. The decrease in FWHM peaks suggests that the crystallite and crystal size increases [20,21].

Moreover, X-ray diffraction patterns provide information about the direction of crystal growth. The absolute area ratio between the a-axis peaks (at 31.7° and 32.8°) and c-axis peaks (at 25.7°) is twice as high in the case of Allograft-Katowice than the other investigated materials. It can be postulated that the HAp nanocrystals of Allograft-Katowice are oriented along the a-axis. In the case of Allograft-Biobank, Allograft-Zimmer, Geistlich Bio-Oss, and Processed-Tooth, the HAp crystals are ordered along their c-axis [9].

The unit cell parameters and the crystal size were calculated by the Rietveld refinement. The results are listed in Table 2. All the investigated materials were built with nanosized apatite crystals. The grain size of allografts materials is in the range of 8.0–12.0 nm. The crystal size of Geistlich Bio-Oss and Processed-Tooth is equal to 21.1 nm and 28.0 nm, respectively. Results clearly show the investigated bone-grafting materials are built with nanosized crystals. A significant amount of the amorphous phase–collagen fibers is also reflected in low values of crystallite size.

### 2.5. Thermogravimetric Analysis

The thermal stability of the investigated bone graft materials was determined by thermogravimetric analysis (TGA). The TGA curves are shown in Figure 9. The three weight-loss steps can be distinguished in the curve. The first step around 100 °C was due to the loss of water. The smaller loss was observed for Geistlich Bio-Oss, approximately 2 wt.%. Similar weight loss was detected for Allograft-Katowice and Processed-Tooth, being equal to 4 wt.%. The most mass lost was observed in Allograft-Zimmer and Allograft-Biobank, at approximately 7 wt.%. 

All investigated materials were relatively stable up to the ambient temperature of 270 °C. The second step of the weight loss in the temperature range of 280–600 °C was attributed to the decomposition of the organic phase. The protein molecules are decomposed into CO_2_ and H_2_O. The third weight loss was indicated up to 850 °C. It may be attributed to the thermal disintegration of CO_2_^3−^ groups from hydroxyapatite molecules [11,12].

From the TGA curves, the residual mass was estimated. The smaller weight loss was obtained for Geistlich Bio-Oss—the residual mass was equal to 93.54%. In the case of Processed-Tooth, it was equal to 77.37%. All allogenic materials were characterized by similar values of 69.21%; 66.36%, and 64.46% for Allograft-Katowice, Allograft-Zimmer, and Allograft-Biobank, respectively. 

Based on the differences in the TGA curves, two heat-treating temperatures for future studies were selected. The representative materials were calcinated at the ambient temperatures of 600 °C and 800 °C. 

### 2.6. Heat-Treating

The XRD patterns were collected for the representatives of the investigated materials after heat-treating. For the analysis, Allograft-Biobank, xenograft Geistlich Bio-Oss, and Processed-Tooth were chosen. In Figure 10, the XRD patterns of the samples heat-treated at 600 °C and at 800 °C are compared with the previously shown patterns of the as-prepared samples. 

On the XRD patterns of the as-prepared Allograft-Biobank, wide lines corresponding with hydroxyapatite nanocrystals were recorded. Two XRD patterns after heat-treating were composed of a series of narrow, sharp lines with similar FWHM. It suggested the growth of apatite crystals after heat-treating already at 600 °C. It should be pointed out that no other crystal phase was found. In the case of Geistlich Bio-Oss, the FWHM was dependent on the heat-treating temperature. The FWHM indicates that the crystal size insignificantly changed after being calcined at 600 °C. Significant crystal size growth was observed after heat-treating at 800 °C, which corresponds with the decrease in the FWHM. In the XRD patterns of the heat-treated samples of Processed-Tooth, the FWHM were changed for both calcinating temperatures, which indicated the increased crystal size. Two extra lines appeared at 2 theta 31.4° and 34.7°. The new lines were ascribed to the β-TCP presence [13,14].

## 3. Discussion

A physical-chemical study shows the differences in the microstructure and nanostructure of bone grafts materials. Based on the SEM images, it was observed that the origin of the material has an influence on the collagen fiber compact level and porosity. Free collagen fibers were detected and recognized in the case of Allograft-Katowice, as well as Allograft-Biobank. The surface of Allograft-Zimmer is a compact, spongy structure. A highly porous surface was found on some parts of Processed-Tooth. Geistlich Bio-Oss presents the rough surface, on which aggregated particles were specified. 

The elements content analysis provided information about minor and mirror elements. The concentration of calcium, phosphate, and oxygen atoms remained in agreement with expectations concerning the presence of hydroxyapatite crystals. The EDS analysis of allograft materials showed significant amounts of carbon in those materials. The elements content can be compared with the FT-IR spectra. On the FT-IR spectra, the carbon atoms were detected in the form of CO_2_^3−^ groups. It can be confirmed that the allograft materials were made of carbonated hydroxyapatite. 

The FT-IR spectra were compared with the XRD patterns. Based on this analysis, it can be postulated that Allograft-Biobank and Allograft-Zimmer included the most organic phases in comparison to the rest of the investigated materials. The protein molecules were also found in the Allograft-Katowice material. In the case of Processed-Tooth and Geistlich Bio-Oss, the presence of organic phase was not confirmed.

The TGA analysis was used to approximate the ratio between the organic and inorganic phases. It was found that the water concentration of the investigated materials varied. Furthermore, the representative of the xenograft material (Geistlich Bio-Oss) was characterized by high thermal stability and the highest concentration of inorganic phase—hydroxyapatite. Thermal stability was confirmed by the XRD measurements, using the sample after heat-treating. The highest amount of organic phase was detected in the case of allograft materials. Independent of origin, the obtained residual mass was similar for all three allograft samples.

The discussed results were the preliminary study for future research. Based on this knowledge, future clinical research will discuss the differences in the bone regeneration process as a function of physic-chemical properties of bone-grafting materials. 

## 4. Materials and Methods

### 4.1. Materials Preparation

#### 4.1.1. Allogenic Bone Block from Zimmer^®^

The material was delivered in sterilized ready-to-use packages in the form of a cortico-cancellous bone plate. The bone was harvested from living donors. The allograft was processed by Zimmer^®^’s Tuttoplast (München, Germany) process procedures, which obtained a sterile, bacteria-free bone substitute with preserved natural bone matrix collagen [15]. Before the investigation, the material was formed into irregular, small pieces.

#### 4.1.2. Allogenic Bone Block from the Tissue Bank in Katowice, Poland

This bone block, contrary to the one produced by Zimmer^®^, was obtained from dead donors and was delivered in sterilized ready-to-use packages; however, its bone type differed. Polish allograft was delivered in the form of a cancellous bone block. From the oral surgery point of view, the material offered other properties; for example, it was more fragile. The material was cut into 4 mm pieces before the evaluations. 

#### 4.1.3. Allogenic Bone Substitute Biobank Global-D

The material was obtained from living donors, after hip arthroplasty. The bone was sterilized in the Supercrit^®^ technological procedure, with CO_2_ used in the liquid state. Moreover, at the end of the preparations, the bone substitute was radiated by gamma radiation of 25 kGy. It was delivered in the most commonly used form, i.e., granules of two different sizes (S- Granulometry 0.5 mm; L- Granulometry 1 mm) packed into small jars. The material was already sterile and could be used right away [16]. 

#### 4.1.4. Bone Substitute Geistlich Bio-Oss^®^

Geistlich Bio-Oss is a representative of a xenogenic bovine origin material. The main advantage of this material was its slow resorption time, which obtained adequate volumes of the newly formatted bone. It looked quite like the previously mentioned granulated material, and comes in small (up to 1 mm particles) and large (particles 1–2 mm) granules. 

#### 4.1.5. Processed Human Extracted Tooth

The extracted tooth was adjusted, dried, and grinded in a Smart Dentin Grinder (Kometabio, Fort Lee, NJ, USA). The manufacturer of Kometabio informed that the size of the obtained particles was between 300–1200 μm in diameter. The particulate tooth was submerged in alcohol cleanser in a sterile container for 7 min to dissolve all organic components and bacteria. Then, the granules were washed with sterile saline for 3 min and dried on medical gauze [23]. 

### 4.2. Physical-Chemical Characterization

The SEM images and the EDS analysis were checked using a Scanning Electron Microscope—Leo Zeiss 435 VP SEM (Zeiss, Oberkochen, Germany), operating at 20 kV. The SEM microscope was equipped with a RONTEC energy dispersive X-ray system. Before the observation, all the materials were gold-sputtered using ScanCoat six equipment—Oxford. 

The Fourier-transformed infrared-spectra were collected by Thermo Scientific Nicolet iS50 FT-IR spectrometer (ThermoFisher Scientific, Waltham, MA, United States) with built-in Automated Beamsplitter exchange system (iS50 ABX containing DLaTGS KBr detector), equipped with all-reflective diamond ATR module (iS50 ATR). The spectra were measured in the range of 4000–400 cm^−1^, mid-infrared region. 

The X-ray diffraction patterns were collected by X’Pert Pro PANalytical diffractometer (Cu, *K_α1_* = 1.54060 Å) (Malvern Panalytical Ltd., Malvern, UK) in the range of a 2θ between 10° to 70°. The X-ray data were analyzed using Match! Software 3.11.1.183. The data values were normalized from 0 to 1. The direction preference of crystal growth was estimated as the ratio of the absolute area below the peaks at 31.7° and 32.8° associated with the a-axis, and the peaks at 25.7° related to the c-axis. 

In the thermogravimetric analysis, a Netzsch TG 209 F1 (Erich NETZSCH GmbH & Co., Holding KG, Selb, Germany) apparatus was used. The measurements were performed at ambient temperature in the range of 30–800 °C, with a heating rate of 10 °C/min, under a nitrogen atmosphere. Before the analysis, the equipment was temperature-calibrated by analyzing six purity standards (tin, indium, zinc, aluminum, bismuth, and silver). For the measurement, approximately 10 mg of a sample was placed in a ceramic vessel. Exactly before each measurement, the analysis of an empty vessel was determined, and the resulting background was subtracted from each thermogram to correct for the equipment drift. The amount of inorganic phase was evaluated as residual mass *m_R_*.

## 5. Conclusions

After tooth loss, and the trauma or inflammation, there is a high possibility of irreversible bone resorption. Without a specific treatment focused on the reconstruction or regeneration of adequate bone volume, the bone will not be a suitable substrate for successful dental treatment. On the current market, numerous materials and procedures for bone reconstruction are available. It is important to choose the best one for each individual case. 

Our study shows that the origin of bone grafting entails different ratios of organic and inorganic phases and water concentration. Additionally, the presence of minor elements like carbon, sodium, or magnesium are dependent on origin. Furthermore, the inorganic phase is built by hydroxy- or carbonate-apatite and characterized by various collagen fiber packing. In the future, we would like to continue this work and present the differences in the rebuilding process based on the CBCT analysis after a surgical procedure. 

## Figures and Tables

**Figure 1 nanomaterials-12-03752-f001:**
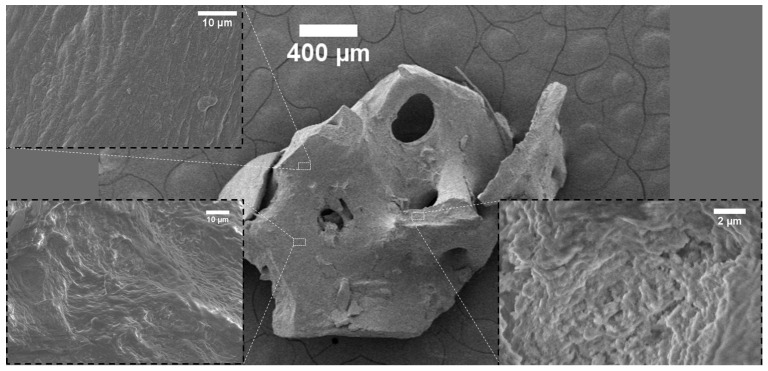
SEM images of Allograft-Zimmer.

**Figure 2 nanomaterials-12-03752-f002:**
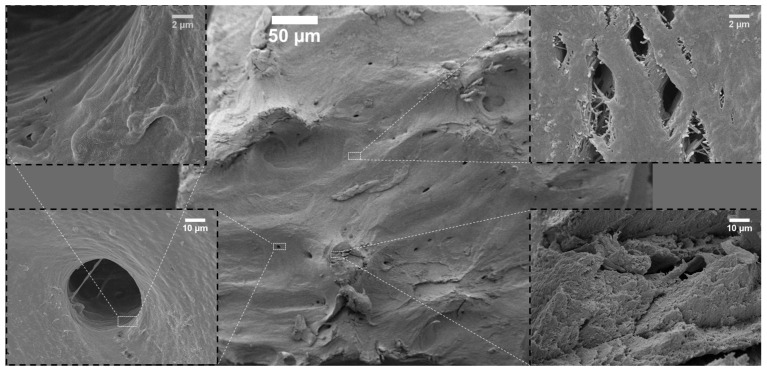
SEM images of Allograft-Katowice.

**Figure 3 nanomaterials-12-03752-f003:**
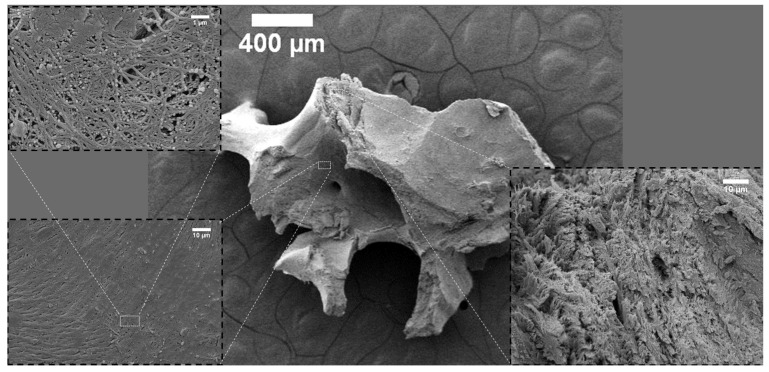
SEM images of Allograft-Biobank.

**Figure 4 nanomaterials-12-03752-f004:**
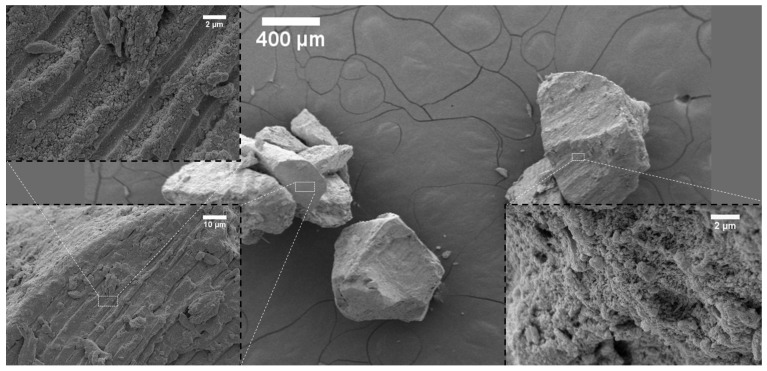
SEM images of Processed-Tooth.

**Figure 5 nanomaterials-12-03752-f005:**
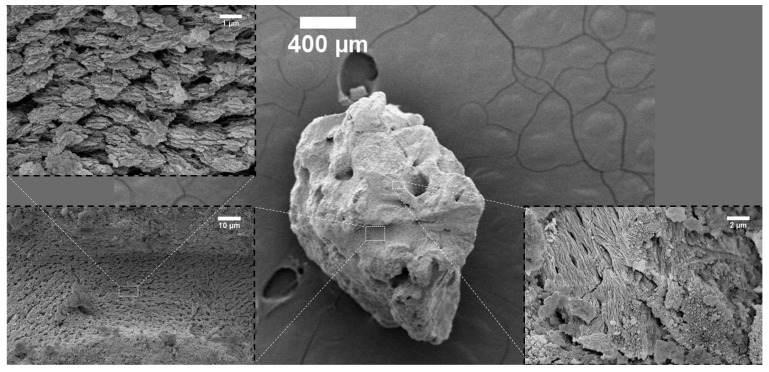
SEM images of Geistlich Bio-Oss.

**Figure 6 nanomaterials-12-03752-f006:**
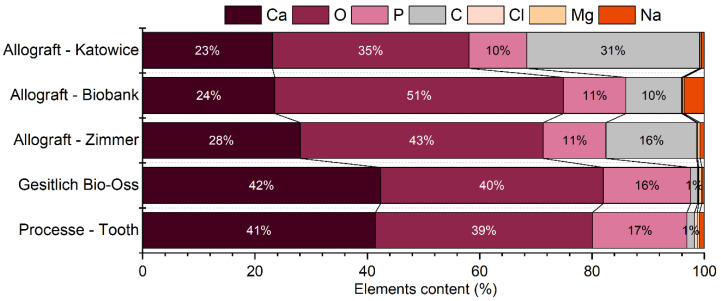
Chemical elements content of investigated materials.

**Figure 7 nanomaterials-12-03752-f007:**
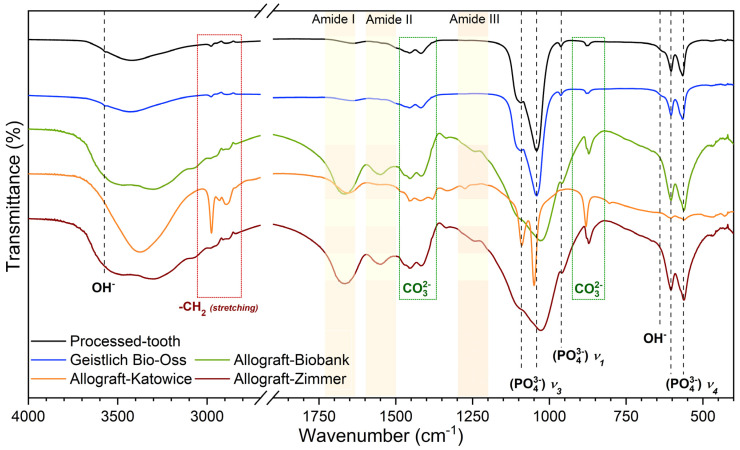
FT-IR spectra of investigated materials.

**Figure 8 nanomaterials-12-03752-f008:**
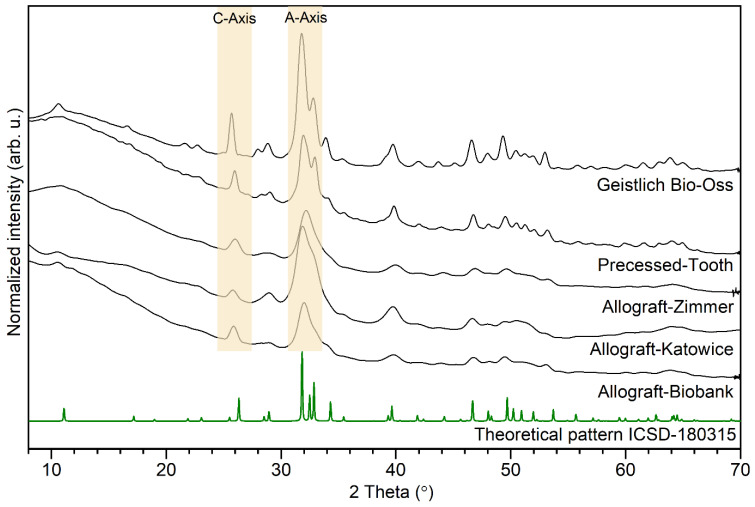
XRD patterns of investigated bone-reconstruction materials.

**Figure 9 nanomaterials-12-03752-f009:**
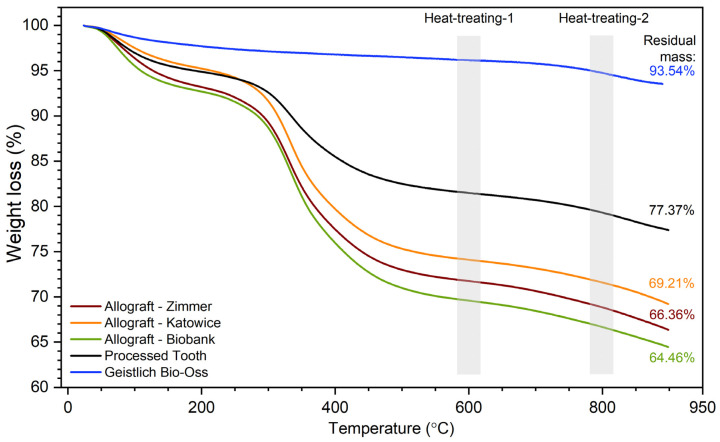
TGA curves of bone graft materials.

**Figure 10 nanomaterials-12-03752-f010:**
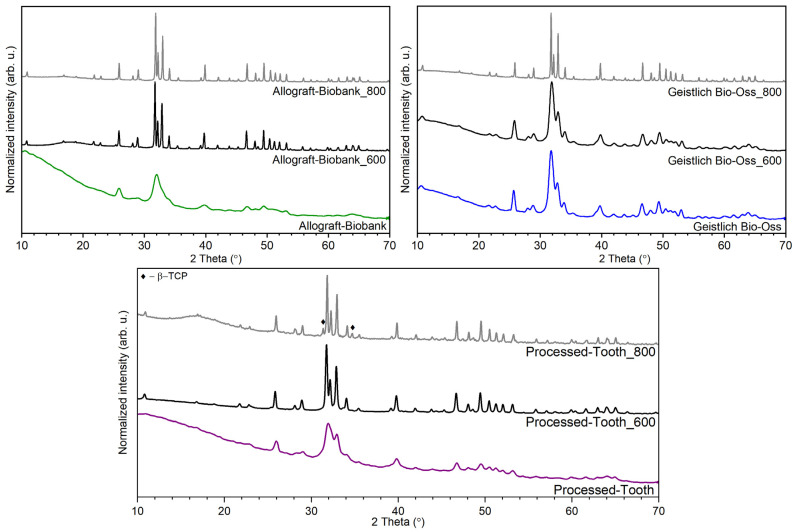
XRD patterns of heat-treated bone-reconstruction materials.

**Table 1 nanomaterials-12-03752-t001:** The chemical composition of investigated bone-replacing materials (EDS analysis).

Elements	Processe–Tooth	Gesitlich Bio-Oss	Allograft–Zimmer	Allograft–Biobank	Allograft–Katowice
C Atom (mol%)	1.36	1.25	16.15	9.93	30.77
O Atom (mol%)	38.68	39.63	43.27	51.38	35.02
P Atom (mol%)	16.78	15.55	11.16	11.11	10.26
Cl Atom (mol%)	0.50	0.19	0.16	0.15	0.10
Ca Atom (mol%)	41.45	42.39	28.06	23.55	23.11
Mg Atom (mol%)	0.39	0.58	0.45	0.27	0.23
Na Atom (mol%)	0.85	0.42	0.75	3.62	0.52

**Table 2 nanomaterials-12-03752-t002:** Unit cell parameters (a,c), cell volume (V), grain size, as well as refine factor (R_W_) for the bone grafting materials.

Sample	a (Å)	c (Å)	V (Å^3^)	size (nm)	R_w_ (%)
single crystal *	9.363 (2)	6.878 (2)	522.18 (27)	–	–
Allograft–Zimmer	9.426 (2)	6.908 (0)	531.56 (49)	12.0 (1)	2.1
Allograft–Biobank	9.414 (4)	6.898 (6)	529.51 (33)	9.4 (3)	2.2
Geistlich Bio-Oss	9.409 (3)	6.894 (3)	528.61 (01)	21.1 (8)	2.9
Allograft–Katowice	9.422 (2)	6.890 (9)	529.79 (91)	8.0 (2)	2.3
Processed–tooth	9.447 (2)	6.899 (6)	533.28 (67)	28.0 (2)	2.7

* Reference to [22].

## Data Availability

Not applicable.

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
