# Peer review of "Investigation of Different Types of Micro- and Nanostructured Materials for Bone Grafting Application"

_nanomaterials, 2022, doi:10.3390/nano12213752_

Round 1

Reviewer 1 Report

Thank you for the opportunity to review this manuscript. 

The paper entitled “Investigation of Different Types of Micro- and Nanostructured Materials for Bone Grafting Application.” is interesting. 

 I would like to accept it in present form.

Author Response

We would like to express our sincerest gratitude for your efforts put into reading the manuscript.

Reviewer 2 Report

Manuscript “Investigation of Different Types of Micro- and Nanostructured Materials for Bone Grafting Application” represents a contribution to field of materials sciences.

In order for the manuscript to represent a contribution to nanomaterials science, it is necessary:

  • In accordance with the title “Investigation of Different Types of Micro- and NANOstructured Materials for Bone Grafting Application”, in the manuscript it is not clear what that NANO is. It is necessary to emphasize the nanostructures in the manuscript. It is necessary to present the results of TEM or FESEM analysis; what is the size from 1 to 100 nm?
  • Within chapter 2.4, it is necessary to show/calculate the crystallite sizes (by Scherrer formula).
  • Line 235 “HAp nanocrystals”? It is necessary to define the size.
  • Based on the results shown in Figure 10, calculate the crystallite size and present it in a table.

Author Response

  1. In accordance with the title “Investigation of Different Types of Micro- and NANOstructured Materials for Bone Grafting Application”, in the manuscript it is not clear what that NANO is. It is necessary to emphasize the nanostructures in the manuscript. It is necessary to present the results of TEM or FESEM analysis; what is the size from 1 to 100 nm?

Answer: We are grateful for the suggestion. Indeed, the TEM images will provide new information. In future study we will try to add also the TEM imaging. For now, it is not possible to perform the TEM imaging. Furthermore, the investigated materials are built by nanosized apatite crystals. It was confirmed by the XRD measurements and crystal size calculation. Therefore, we would like to point it out in the title of the article.

  1. Within chapter 2.4, it is necessary to show/calculate the crystallite sizes (by Scherrer formula).

Answer: We would like to thank to the Reviewer for this sufficient suggestion. The crystallite size, and unit cell parameters were calculated by the Rietveld method. The results are listed in the Table A1 and added into main text.

Table A1. Unit cell parameters (a,c), cell volume (V), grain size as well as refine factor (RW) for the bone grafting materials.

Sample

a (Å)

c (Å)

V (Å3)

size (nm)

Rw (%)

single crystal*

9.363(2)

6.878(2)

522.18(27)

Allograft - Zimmer

9.426(2)

6.908(0)

531.56(49)

12.0(1)

2.1

Allograft - Biobank

9.414(4)

6.898(6)

529.51(33)

9.4(3)

2.2

Geistlich Bio-Oss

9.409(3)

6.894(3)

528.61(01)

21.1(8)

2.9

Allograft - Katowice

9.422(2)

6.890(9)

529.79(91)

8.0(2)

2.3

Processe - tooth

9.447(2)

6.899(6)

533.28(67) 

28.0(2)

2.7

  1. Line 235 “HAp nanocrystals”? It is necessary to define the size.

Answer: Based on the previous answer it was shown the crystal size of each sample. It was confirmed the nanosized of apatite crystals presented in investigated materials.

  1. Based on the results shown in Figure 10, calculate the crystallite size and present it in a table.

Answer: We would like to thank for the suggestion. The crystal size and unit cell parameters are calculated and presented in the Table 2.

Reviewer 3 Report

Revision AR

Comments to the Author

The present study reported on “Investigation of Different Types of Micro- and Nanostructured 2 Materials for Bone Grafting Application”The aim of the study is clear, the methodology appropriate, the data set relevant and the results clearly reported. It is suitable for publication within this Journal after minor corrections:

1. Abstract: OK

2. Introduction: please add the following updated references after “The inadequate bone volume of the alveolar process of the maxilla and the alveolar 26 part of the mandible can be found in the majority of patients undergoing various types of dental treatment.” (PMID: 33548077; PMID: 31927693). Moreover a general sentence on the reliability of dental implants in the long-term in needed especially in light of several recent long-term data (PMID: 32991763; PMID: 33448601; PMID: 36054302; PMID: 35103325)

3. Materials & Methods: OK

2. Discussion: please, add a sentence within this section to underline the need of further research which might have a clinical impact in the future.

Author Response

  1. Abstract: OK

Answer: We would like to express our sincerest gratitude for your efforts put into reading the manuscript.

  1. Introduction: please add the following updated references after “The inadequate bone volume of the alveolar process of the maxilla and the alveolar 26 part of the mandible can be found in the majority of patients undergoing various types of dental treatment.” (PMID: 33548077; PMID: 31927693). Moreover, a general sentence on the reliability of dental implants in the long-term in needed especially in light of several recent long-term data (PMID: 32991763; PMID: 33448601; PMID: 36054302; PMID: 35103325)

Answer: We would like to thank for the suggestion. The main text was corrected, and all changes are marked with the red color (lines 26-28 and 54-58).

  1. Materials & Methods: OK

Answer: Thank you Reviewer.

  1. Discussion: please, add a sentence within this section to underline the need of further research which might have a clinical impact in the future.

Answer: We would like to thank for the suggestion. The extra sentence was added into main text and all changes are marked with the red color (line 322-324)

Round 2

Reviewer 2 Report

The authors corrected the manuscript in accordance with the suggestions of the reviewer.
I recommend accepting the manuscript.